# Regulation of Wound Healing by the NRF2 Transcription Factor—More Than Cytoprotection

**DOI:** 10.3390/ijms20163856

**Published:** 2019-08-08

**Authors:** Paul Hiebert, Sabine Werner

**Affiliations:** Institute for Molecular Health Sciences, Department of Biology, Swiss Federal Institute of Technology Zürich, 8093 Zurich, Switzerland

**Keywords:** NRF2, wound healing, keratinocyte, fibroblast, inflammation

## Abstract

The nuclear factor-erythroid 2-related factor 2 (NRF2) transcription factor plays a central role in mediating the cellular stress response. Due to their antioxidant properties, compounds activating NRF2 have received much attention as potential medications for disease prevention, or even for therapy. Accumulating evidence suggests that activation of the NRF2 pathway also has a major impact on wound healing and may be beneficial in the treatment of chronic wounds, which remain a considerable health and economic burden. While NRF2 activation indeed shows promise, important considerations need to be made in light of corresponding evidence that also points towards pro-tumorigenic effects of NRF2. In this review, we discuss the evidence to date, highlighting recent advances using gain- and loss-of-function animal models and how these data fit with observations in humans.

## 1. The Wound Healing Process

As the organ continuously exposed to the outside world, the skin is charged with many important functions, including maintaining body temperature, hydration, and sensing the environment. Among its most important functions is to provide a barrier to an external environment rich in pathogens and toxins. Furthermore, it is frequently exposed to harmful ultraviolet (UV) radiation. Given the importance of maintaining the integrity of the skin, mammals have evolved a complex and sophisticated wound healing response, whereby damaged tissue is efficiently repaired following injury. While the complex nature of this response contributes to its effectiveness, it also leads to challenges when attempting to remedy pathological scenarios where wound healing goes wrong, such as in chronic, non-healing wounds/ulcers and in hypertrophic scars and keloids [1,2].

The normal wound healing response involves a series of overlapping phases beginning with haemostasis and the formation of a fibrin clot. Haemostasis is initiated by platelets immediately after injury and functions critically to reduce blood loss from damaged vessels. It results in the deposition of a provisional matrix made primarily of fibrin that provides a rapid seal of the wound. This temporary fibrin mesh also functions as a scaffold for cells to migrate into the wound and as a reservoir of cytokines and growth factors. The inflammatory phase follows soon after, with neutrophils being the first immune cells to arrive at the wound site due to their abundance in circulation. They function as an innate response to invading pathogens through their production of proteinases and reactive oxygen species (ROS), which, however, can also damage the cells at the wound site. In addition, neutrophils assist in the recruitment of other immune cell types, such as macrophages and lymphocytes, which accumulate in the following days. Inflammatory cells also function critically to produce cytokines and growth factors, which serve to attract fibroblasts, stimulate keratinocyte migration and proliferation, and initiate the formation of new blood vessels [3,4]. This provides a ‘kick-start’ for the tissue formation phase of wound healing, where keratinocytes at the wound edge begin to migrate into the wound along the injured dermis and then across the provisional wound matrix. Keratinocytes behind the migrating tongue accelerate their proliferation rate to produce a pool of new keratinocytes that replenish those that are lost by the wound [5]. Slightly after the onset of re-epithelialization, dermal fibroblasts at the wound edge migrate into the wound bed, where they proliferate and produce large amounts of extracellular matrix (ECM). A large percentage of the wound fibroblasts differentiate into myofibroblasts, which strongly contribute to wound contraction [6]. Additional sources of wound (myo) fibroblasts are bone marrow-derived cells and adipocyte precursors, which differentiate into (myo) fibroblasts at the wound site [7,8,9]. In parallel, endothelial sprouting occurs at the wound edge, and massive angiogenesis leads to the formation of a new vasculature in the newly forming stromal part of the wound, which is called granulation tissue [1,2]. Once keratinocytes have fully covered and closed the wound, fibroblasts and immune cells continue to remodel the underlying tissue, replacing the provisional matrix with one composed primarily of collagen type I. This remodeling phase also features increased apoptosis of immune cells, fibroblasts and endothelial cells of vessels that do not contribute to the vasculature of the mature scar tissue. Fibroblasts that remain in the scar tissue can continue to remodel the healed skin for up to several years. While the resulting scar lacks the full mechanical strength and elasticity of the original skin and also all appendages, it functions effectively as a restored skin barrier that once again offers protection to environmental dangers [1,2,10,11]. The wound healing response, therefore, involves many carefully orchestrated events that need to provide the right signals to the right cells at the right time. Defects in any one of these wound healing phases can have disastrous consequences and lead to chronic, non-healing wounds followed by infection, hospitalization, or even death [12,13]. Alternatively, healing can be excessive, resulting in the formation of hypertrophic scars and keloids, which can cause severe functional and cosmetic impairments [14,15].

The complex gene expression pattern during wound healing is controlled by various key transcription factors, which orchestrate different processes at the wound site [16,17]. One such transcription factor, mainly known for its cytoprotective activities, is NRF2, which will be introduced below.

## 2. The Transcription Factor NRF2

Nuclear factor-erythroid 2-related factor 2 (NRF2; NFE2L2) is a member of the cap‘n‘collar family of transcription factors, which also includes NFE2, NRF1 (NFE2L1), NRF3 (NFE2L3), BACH1, and BACH2 [18]. NRF2 is particularly famous for its central role in regulating the transcription of cytoprotective genes, including genes encoding various ROS-detoxifying enzymes, antioxidant proteins, and drug transporters [18,19,20]. Over the last three decades, research has demonstrated the remarkable importance of NRF2 in maintaining the cellular redox status in virtually all organs and tissues. Under homeostatic conditions, when only low amounts of the cytoprotective target proteins are needed, NRF2-mediated transcription is restricted to baseline levels by the NRF2 antagonist Kelch-like ECH-associated protein 1 (KEAP1). Binding of NRF2 to KEAP1 sequesters it in the cytoplasm and leads to its rapid proteasomal degradation. The presence of ROS and/or electrophilic compounds weakens the NRF2-KEAP1 interaction, allowing NRF2 to become stabilized. This results in accumulation of newly produced NRF2 in the nucleus. NRF2 acts by dimerizing with small musculoaponeurotic fibrosarcoma (MAF) proteins and binding to specific regions of DNA called antioxidant response elements (AREs), where it initiates the transcription of its target genes [18,21] (Figure 1). In this way, the NRF2 antioxidant pathway serves to protect cells from dangerous levels of ROS that are generated in response to environmental stressors (e.g., UV irradiation, pathogens, and toxins) and during excessive inflammation. However, multiple studies identified additional NRF2 targets that are not directly involved in the antioxidant response and many of them are expressed in a tissue-specific and situation-dependent manner [22,23,24,25]. This is also relevant for wound healing and will be described in detail below.

The benefits of activating the NRF2 pathway have generated intense interest in recent years, and NRF2 activating compounds continue to be tested for their health benefits and prevention of disease [26,27,28]. However, the cytoprotective benefits of NRF2 have been met with increased caution as evidence continues to accumulate suggesting a detrimental role in cancer, where NRF2-mediated protection of cancer cells can contribute to accelerated tumor growth and to chemo- and radio-resistance. Examples like these point to both a good and bad side of NRF2 in health and disease [27,28].

## 3. Expression and Activity of NRF2 in Healing Wounds

Wounds produce large amounts of ROS to combat invading pathogens [16], to attract immune cells, and to regulate different cell signaling events [29]. As such, the role of NRF2 during the wound healing process has been of interest to many researchers. Several lines of evidence suggest an important role for NRF2 during tissue repair. This includes the observation that *Nrf2* gene expression increases early after full-thickness wounding of mice along with ROS production, followed by a gradual reduction as wound healing progresses [30]. Keratinocytes of the hyperproliferative epithelium of skin wounds were shown to strongly express *Nrf2*, but expression of this gene was also seen in cells of the granulation tissue [30]. NRF2 has also been shown to become activated after tissue damage and possibly synergizes with other transcription factors to promote wound repair [31].

A number of studies have tested NRF2-activating compounds for their impact on wound healing and have shown interesting results. These include purified molecules (including nitric oxide (NO) or NO donors), as well as bee venom, olive oil, and various plant extracts that include these compounds [32,33,34,35,36,37,38]. However, it is important to note that these chemicals often have multiple effects and modes of action, making the precise role of NRF2 in these studies unclear. Genetic approaches in mice using NRF2 gain- and loss-of-function models offer an alternative strategy to investigating NRF2 function during tissue repair. Recent studies employing this strategy have added to the evidence for NRF2 as a key player in the wound healing process, with actions extending beyond its antioxidant effects. However, as is typical for NRF2, this evidence occasionally offers a mixed view of both the benefits and possible detriments of activating the NRF2 pathway. In this review, we will discuss the accumulating evidence suggesting an important role for NRF2 during cutaneous wound healing, focusing on findings from genetic animal models, but extending to the potential use of chemical activators of the NRF2 pathway for the promotion of wound healing in humans. 

## 4. Consequences of NRF2 Loss-of-Function during Wound Healing

Global NRF2 knockout (KO) mice have no obvious skin phenotype under normal conditions and were first used in wound healing studies nearly two decades ago. Surprisingly, NRF2 KO mice healed wounds at normal rates and displayed no obvious histological differences compared to control animals [30]. Similar rates of cell proliferation and apoptosis between NRF2 KO and wild-type control mice were also observed in this study. Loss of NRF2 did, however, lead to a delayed induction, but prolonged overexpression of pro-inflammatory cytokines at the wound site and persistence of macrophages, even after closure of the wounds. Among the affected cytokines was transforming growth factor β 1 (TGF-β1), whose delay at the beginning of wound healing may also lead to the reduced collagen production observed in wounds of NRF2 KO mice [30]. 

Despite the observed differences in cytokine production, the lack of a difference in wound healing kinetics in NRF2 KO vs. control mice may be due, at least in part, to compensatory actions of other transcription factors (e.g., NRF1 or NRF3) [30]. To address the issue of compensation, mice were generated that express a dominant-negative NRF2 mutant (dnNRF2) in keratinocytes [39]. The dnNRF2 mutant lacks the KEAP1 binding domain and the transactivation domains, leading to the nuclear accumulation of dnNRF2 that binds to AREs without facilitating transcription. In doing so, other cap‘n‘collar transcription factors are prevented from binding to these AREs, thereby avoiding potential compensatory effects caused by a loss of NRF2. Similar to global NRF2 KO mice, however, keratinocyte-specific dnNRF2 mice showed no differences either in normal skin morphology or during wound healing [39]. Furthermore, mice lacking either NRF2 in keratinocytes or NRF3 in all cells healed normally [40,41]. This was surprising given previous findings showing NRF2 and NRF3 predominantly and highly expressed in the hyperproliferative wound epithelium [30]. However, keratinocytes possess multiple, and partially redundant antioxidant defense pathways [40,42], and loss of NRF2 activity may be compensated by other cytoprotective factors and pathways, allowing the epithelium to properly regenerate.

It is likely that certain actions of NRF2 differ substantially depending on cell type, as cells located in different organs and tissues have diverse functions, specific NRF2 targets [22,23], and are exposed to varying degrees of environmental stressors. Skin fibroblasts for example, are located within the dermis and are shielded from many of the environmental dangers faced by keratinocytes. In vitro, skin fibroblasts deficient in NRF2 showed slightly increased levels of ROS, however, proliferation was similar to control cells [43]. In line with other NRF2 loss-of-function studies, mice with fibroblast-specific deletion of NRF2 also showed no differences in wound healing kinetics compared to control animals [43]. These mice did feature an altered immune cell content in unwounded skin, but these differences disappeared upon wounding [43]. Similar to global NRF2 KO mice, this suggests that NRF2 may have some influence over immune cell recruitment, however, the effect may be diluted in the context of a healing wound, where many other immune-regulatory mechanisms are at play. 

The prolonged wound inflammation in NRF2-deficient mice suggested a possible role for NRF2 in immune cells. Immune cells of the myeloid lineage have important functions during wound healing in fighting off invading bacteria through the production of ROS and in the production of healing-promoting factors [3]. In light of these observations, mice lacking NRF2 in myeloid cells were generated and subjected to full-thickness excisional wounding [44]. In wounds of wild-type mice, macrophages, and to an even greater extent neutrophils, were shown to strongly express *Nrf2* and cytoprotective NRF2 target genes (e.g., *Nqo1*). Nevertheless, and similar to other cell types, myeloid-specific NRF2 KO mice showed no difference in wound healing as revealed by morphometric analysis of histological wound sections and analysis of the wound immune cell content [44]. One caveat to consider is that wound healing studies conducted on laboratory mice are typically performed in very clean conditions. In situations that feature greater exposure to an abundant variety of bacterial or other environmental challenges, it is possible that NRF2 deficiency would prove to be more detrimental. Furthermore, the majority of these studies have focused primarily on examination of wound closure with, to date, less attention given to the effect of NRF2 deficiency on scarring. Future work will likely address these open questions further, however the bulk of the evidence thus far strongly suggests that, while NRF2 may have important roles to play, its function is often redundant, making it dispensable for wound healing in healthy mice that do not face additional challenges.

In contrast to normal wound healing, pathological wound healing represents a situation where the normal wound healing process has failed, leading to both chronic wounds, ulcers, or hypertrophic scars and keloids. Due to the increased challenge faced by the skin in these scenarios [1,2], it is possible that the loss of NRF2 would have a more important impact. This hypothesis is supported by experiments performed using a diabetic mouse model of delayed wound healing [45]. Mice injected with streptozotocin develop type I diabetes and exhibit significantly delayed wound closure compared to control mice. Interestingly, wound closure was delayed even further in streptozotocin-induced diabetic NRF2 KO mice [45], suggesting that unlike normal wound healing, NRF2 may indeed have a more crucial role in facilitating wound closure in pathological situations, such as diabetes. This hypothesis is further supported by the finding that the NRF2 pathway is functionally insufficient in bone marrow-derived multipotent stromal cells from diabetic mice [46]. The consequences of NRF2 loss- or gain-of-function on the wound healing process are summarized in Table 1.

## 5. Consequences of NRF2 Activation during Wound Healing

Typically, it is in pathological scenarios where NRF2 activators continue to gain attention as possible therapeutics due to their antioxidant properties. As shown in diabetic mice, NRF2 activity appears to be insufficient during impaired wound healing, resulting in excessive ROS levels, which can lead to severe tissue damage and uncontrolled inflammation. Consistent with this assumption, delayed wound healing observed in mice that are diabetic as a consequence of a mutation in the leptin receptor (Lepr*^db/db^* mice) and in streptozotocin-induced diabetic mice was significantly improved through the use of NRF2 activating compounds when applied either systemically or topically [32,45]. Administration of such compounds also led to the restoration of normal TGF-β1 and matrix metalloproteinase (MMP)9 levels in the skin and reduced the oxidative burden [32,45]. Furthermore, NRF2 activating compounds promoted proliferation and migration of the immortalized HaCaT keratinocyte cell line in vitro under hyperglycaemic conditions [45]. It has also been shown that drugs used to treat diabetes can activate the NRF2 pathway [48]. Interestingly, these same drugs have shown efficacy in promoting diabetic wound healing, although in this instance, activation of the NRF2 pathway may not be the principle mechanism of action [49]. Taken together, these data point to activation of the NRF2 pathway as a promising strategy to promote wound healing under stress conditions (Figure 2).

As mentioned, one downside of using NRF2 activators to elucidate the role of NRF2 in vivo is their likely multiple modes of action, which make the precise role of NRF2 difficult to determine. Alternatively, genetic approaches in rodents have been used to study the consequences of NRF2 activation. In one study, exosomes produced by adipose-derived stem cells overexpressing NRF2 were shown to improve the ulceration of foot wounds in diabetic rats [50]. Furthermore, closure of ear holes was accelerated in mice with a gain of function mutation in the gene encoding rhomboid family protein RHBDF2, and this was associated with activation of the NRF2 pathway [51]. However, the specific contribution of NRF2 in the beneficial effect of RHBDF2 is not fully clear. To more specifically address the effect of activated NRF2 in vivo, KEAP1 KO mice were generated [52]. As the principle NRF2 antagonist, KEAP1 deletion results in constitutive activation of NRF2 in the cell without the need for any additional stimulus. However, global KEAP1 KO mice do not survive longer than 2 to 3 weeks, most likely due to irregular cornification and hyperkeratosis in the esophagus, resulting in gastric obstruction [52]. When diabetic mice were wounded and treated with KEAP1 siRNA, wound healing was significantly improved, featuring faster wound closure, elevated granulation tissue formation, and reduced ROS-mediated formation of 8-oxo-2’-deoxyguanosine [53,54]. Furthermore, NRF2 activation in bone marrow-derived multipotent stromal cells of diabetic mice through knock-down of KEAP1 restored their multipotent cell properties and promoted wound healing in diabetic mice [46].

Due to the early lethality of KEAP1 KO mice, and because KEAP1 also targets other factors in the cell besides NRF2 [55,56,57], an alternative approach for achieving constitutive NRF2 activation is cell-specific expression of a constitutively active NRF2 (caNRF2) mutant [58,59]. The caNRF2 mutant lacks the KEAP1 binding domain and therefore remains constitutively active in the nucleus without affecting interactions between KEAP1 and other proteins. 

Mice expressing low levels of caNRF2 in keratinocytes only exhibit mild hyperkeratosis in the tail skin, but they are protected from UVB-induced cell death [58]. However, stronger expression of caNRF2 that reflects the activation level achieved after activation of endogenous NRF2 by chemical activators, caused obvious skin abnormalities, including mild inflammation, hyperkeratosis, sebaceous gland hypertrophy, and defects in epidermal barrier function [60]. Interestingly, these mice showed enhanced re-epithelialization during wound healing, leading to significantly faster wound closure [47]. While improved wound healing in these mice is in agreement with results observed in diabetic mice treated with NRF2 activating compounds, caNRF2 expression did not lead to differences in keratinocyte migration, and resulted in no change in keratinocyte proliferation in the wound epidermis. Instead, caNRF2 increased proliferation of cells within hair follicles and sebaceous glands at the wound periphery that expressed keratinocyte stem cell markers [47]. Stem cell subpopulations in the hair follicle bulge, the junctional zone and the upper isthmus were previously shown to contribute to wound re-epithelialization [61] and caNRF2 specifically stimulated the proliferation of cells with markers of junctional zone and upper isthmus stem cells [47]. Therefore, the additional pool of cells that are generated at the wound edge most likely via stem cell expansion seems to function as an increased reservoir of cells available to migrate into and re-populate the wound. The expansion of these stem cell populations is plausibly driven by NRF2-mediated expression of epigen in these cells. This epidermal growth factor family member, which is encoded by a direct target gene of NRF2 [60], has been identified as a strong promoter of hair follicle stem cell proliferation in vivo [62]. Interestingly, faster wound closure in caNRF2-transgenic mice required a sufficient degree of NRF2 activation, as mice expressing lower levels of the caNRF2 transgene failed to show differences in wound closure, despite a slight elevation in the expression of NRF2 target genes [47]. These data reinforce the idea that NRF2 activation can have benefits for wound healing and that the degree of NRF2 activation may be important to consider when evaluating its effectiveness.

The consequences of NRF2 activation in fibroblasts have also been investigated. Fibroblasts in the skin play critical roles during wound healing, including the release of cytokines and growth factors, deposition of ECM, and wound contraction [2,63]. Remarkably, activating NRF2 in fibroblasts lead to fibroblast senescence both in vitro and during wound healing in vivo [43]. This was shown to be driven largely by the NRF2-mediated deposition of an altered matrisome featuring elevated levels of the senescence-promoting factor plasminogen activator inhibitor-1 (PAI-1, serpine1). Accelerated senescence was observed both in caNRF2-expressing fibroblasts and in wild-type fibroblasts treated with the NRF2 activator tert-butyl-hydroquinone, and occurred despite reduced levels of intracellular ROS and reduced DNA damage [43]. This is of relevance during tissue repair as senescent cells provide important molecular cues to neighboring cells via a growth-promoting secretome referred to as the senescence-associated secretory phenotype (SASP). Furthermore, senescent cells have been shown to be important for normal wound healing [64]. caNRF2 fibroblasts do indeed display a SASP and release factors capable of directly promoting keratinocyte proliferation [43]. Accordingly, mice expressing caNRF2 in fibroblasts showed significantly faster wound closure, featuring enhanced re-epithelialization as a result of increased keratinocyte proliferation in the wound epidermis.

Unlike keratinocytes and fibroblasts, the impact of NRF2 activation in myeloid cells on wound healing has proven to be relatively little by comparison. Mice expressing caNRF2 in myeloid cells showed no significant difference in wound closure rates and even failed to show increased expression of NRF2 target genes to a significant degree [44]. A likely explanation for this is that these cells, particularly neutrophils, display very high levels of NRF2 expression and activation even during homeostasis and in the circulation, thereby dwarfing any additional effect of the caNRF2 transgene. This stands in contrast to NRF2 activation in myeloid cells in other organs, where genetic deletion of KEAP1 resulted in protective effects in mouse models of sepsis and ischemia/reperfusion injury in the liver [65,66]. Nevertheless, activation of NRF2 in the wound tissue seems to predominantly affect non-immune cells. Taken together, the bulk of the evidence supports activation of NRF2 as a promising strategy to promote wound repair, particularly in situations where healing is negatively affected by enhanced ROS levels (Figure 2).

## 6. Cancer and the Dark Side of NRF2

For many years, researchers have observed important parallels between the wound healing process and cancer [67]. This is also evident when considering the potential benefits of NRF2 on wound healing, actions which may prove unwelcome during tumorigenesis [27,68]. This stands to reason given that activating cellular defense pathways tend to promote cell survival, which is undesirable in tumor cells and in conflict with the actions of chemotherapeutic drugs and with radiotherapy [68]. In humans, multiple lines of evidence support both pro- and anti-tumorigenic roles for NRF2 including during skin cancer development and progression [69], a phenomenon that can also be observed in gain- and loss-of-function animal models. For example, hyperactivation of NRF2 as a result of KEAP1 loss-of-function protected from UV-induced skin carcinogenesis [70,71], while mice expressing low levels of caNRF2 in keratinocytes showed increased tumor incidence and multiplicity in a genetic model of Human Papilloma Virus 8-induced skin cancer, resulting primarily from increased survival of keratinocytes during the early stages of the transformation process [72]. In contrast, these same mice expressing caNRF2 in keratinocytes showed mildly reduced tumor incidence and multiplicity in a chemically-induced model of skin cancer, where NRF2-mediated detoxification of the mutagen and of ROS induced during the treatment overruled the pro-tumorigenic activity of NRF2 [72]. A beneficial aspect of a basal activity of NRF2 in cancer development is supported by experiments performed in mice with NRF2 loss-of-function in keratinocytes. Here, the same chemically-induced skin carcinogenesis protocol lead to significantly greater tumor multiplicity in mice lacking a functional NRF2, suggesting an important role for NRF2 in cancer prevention [39,73]. This speaks to the importance of context, as NRF2 activation in healthy cells will indeed lead to reduced cellular stress, preventing DNA damage and cancer-causing mutations, an action that becomes detrimental, however, once malignant transformation has occurred and once NRF2 becomes hyperactivated. Separate from its classical role in detoxification of ROS, NRF2 can also influence PI3K signaling, and has been shown to activate transcription of the gene coding for mechanistic Target of Rapamycin (mTOR), a common promoter of tumor growth [74]. Interestingly, mTOR signaling may also have important implications for wound healing [75,76,77], however the role of NRF2-mediated mTOR activation in this context remains to be determined.

The pro-tumorigenic actions of NRF2 can also be extended beyond the cancer cells. Mice expressing caNRF2 in fibroblasts have been shown to express a cancer-associated fibroblast (CAF) gene expression signature, capable of promoting tumor growth in vivo in a xenograft model of squamous cell carcinoma (SCC) [43]. Here, SCC cells were co-injected with caNRF2-expressing fibroblasts into the skin of immunocompromised mice. The activated NRF2 induced a CAF phenotype in these cells and led to the SCC cells forming significantly larger tumors compared to those co-injected with control fibroblasts [43]. This suggests that in addition to promoting cancer cell survival, NRF2 activation in other cell types (e.g., CAFs or others) may have additional impacts on cancer development, and that NRF2 activation can have multiple consequences beyond cytoprotection.

## 7. Summary and Future Perspectives

The last three decades have seen an explosion of interest in NRF2 and its antioxidant role, cementing its importance in maintaining the cellular redox status. During this time, many researchers have investigated the use of NRF2 activating compounds as a means of cancer prevention and for preventing, or even treating, chronic inflammatory and degenerative diseases [26,27,28]. However, the antioxidant activity also results in survival of cancer cells under stress conditions and, together with NRF2’s important function in drug detoxification, makes them chemo- and radioresistant [27]. Furthermore, recent advances using genetic animal models have uncovered additional and unexpected consequences of NRF2 activation beyond cytoprotection. In the skin, this included promotion of keratinocyte stem cell proliferation and of fibroblast senescence, impairment of epidermal barrier function, induction of sebaceous gland hyperplasia, and deposition of an abnormal ECM [43,47]. Interestingly, most of these activities are controlled by cell type-specific NRF2 target genes that are not conventional NRF2 targets under homeostatic conditions. 

In addition to the promising effects on wound healing observed in mice, many of these findings may be translated to humans (Figure 2). NRF2 and its target genes were shown to be activated in the hyperthickened epidermis of human diabetic ulcers, likely as a consequence of elevated ROS levels and reflective of the severe oxidative stress that occurs in these wounds [45]. This may well protect keratinocytes from oxidative stress to a certain extent and this protection may be further promoted by application of NRF2 activating compounds. Indeed, and as mentioned above, human keratinocytes (HaCaT cell line) are stimulated to proliferate when treated with NRF2 activators under hyperglycaemic conditions, similar to observations in diabetic mice treated with these same compounds [45]. In the future it will be important to determine if NRF2 is also activated in fibroblasts of chronic human wounds, which could contribute to the fibroblast senescence that is frequently seen in skin ulcers [78]. Indeed, human fibroblasts treated with NRF2 activating compounds, or with CRISPR/Cas-mediated deletion of *KEAP1*, also undergo senescence at an accelerated rate compared to untreated cells [43]. However, it remains to be determined if these cells are still able to produce a healing promoting SASP in chronic wounds/ulcers as seen in acute mouse wounds [43]. For other effects of NRF2 activation in mouse wounds, such as expansion of stem cells within hair follicles [47], the impact on human wounds is unclear, as human skin typically contains far fewer hair follicles than mouse skin. As such, these effects may be more relevant to specific areas of the skin with a higher hair follicle density.

Together, these findings demonstrate the multifaceted actions of NRF2 during tissue repair, and have helped uncover important mechanisms that may benefit humans suffering from chronic wounds. However, it is also evident from multiple studies that pro-tumorigenic actions and other potentially detrimental activities should remain an important consideration prior to employing NRF2 activating compounds to wounds in the clinic. Therefore, application of such compounds should be limited to short treatment periods and restricted to patients in which malignancy at the wound site is excluded. Future work aimed at unravelling beneficial vs. deleterious actions of NRF2 and identification of the relevant target genes and their regulation will be of particular importance. A better understanding of these matters will potentially lead to the development of more potent and safer NRF2 activators that may reliably be used to promote wound healing in patients.

## Figures and Tables

**Figure 1 ijms-20-03856-f001:**
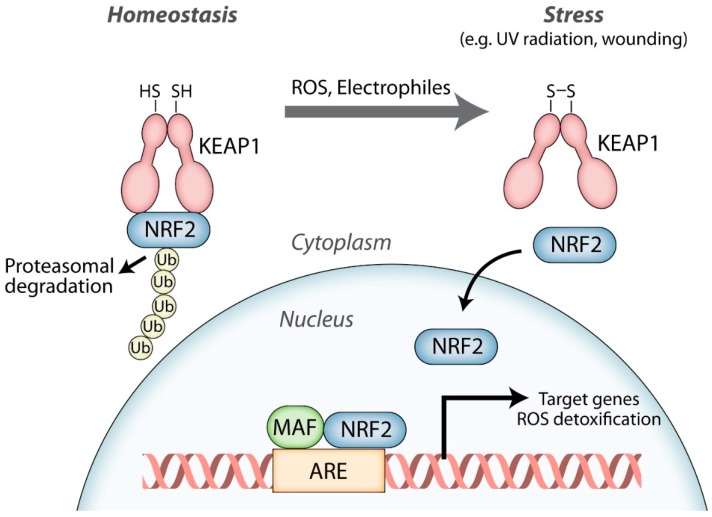
The nuclear factor-erythroid 2-related factor 2 (NRF2) signaling pathway. NRF2 strongly binds to its cytoplasmic inhibitor Kelch-like ECH-associated protein 1 (KEAP1) under homeostatic conditions and only low levels of NRF2 are present in the nucleus. In response to reactive oxygen species (ROS) and/or electrophiles, the NRF2-KEAP1 interaction is weakened and newly formed NRF2 accumulates in the nucleus. Here, NRF2 dimerizes with small musculoaponeurotic fibrosarcoma (MAF) proteins and binds to antioxidant response elements (AREs) in the promoters or enhancers of its target genes, of which many encode ROS detoxifying enzymes and other antioxidant proteins, thereby initiating a cytoprotective response.

**Figure 2 ijms-20-03856-f002:**
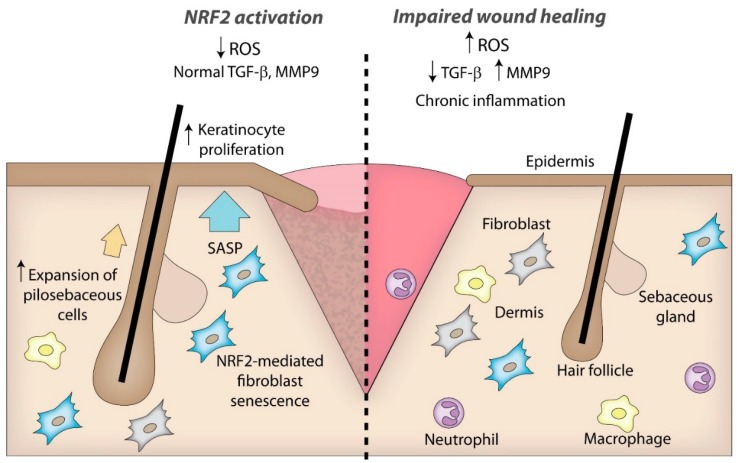
Characteristic features of chronic wounds, which may be improved by treatment with NRF2 activating compounds. NRF2 activation reduces oxidative stress, thereby enhancing production of TGF-β1, which is important for granulation tissue formation and matrix production. Reduction of ROS also suppresses the chronic inflammation and the excessive production of MMP9. Expansion of pilosebaceous cells by activated NRF2 may promote re-epithelialization of the wound. Activation of NRF2 in fibroblasts may promote senescence and associated production of a SASP, which can further promote wound re-epithelialization. Arrows pointing to the top indicate upregulation and arrows pointing to the bottom indicate downregulation.

**Table 1 ijms-20-03856-t001:** NRF2 gain- or loss-of-function phenotypes during wound healing.

Cell Type	Loss-of-Function Phenotype	Gain-of-Function Phenotype
Global	No change in wound closure rates [30] Delayed induction of cytokines [30] Prolonged wound inflammation [30] Reduced collagen deposition [30] Delayed diabetic wound healing [45]	Improved diabetic wound healing [45] Increased keratinocyte proliferation [45] Restoration of normal transforming growth factor β 1 (TGF-β1) and matrix metalloproteinase (MMP)9 levels [45]
Keratinocytes	No change in wound closure rates [39]	Accelerated wound closure/re-epithelialization [47] Increased expansion of pilosebaceous cells [47] No change in keratinocyte proliferation within the wound [47]
Fibroblasts	No change in wound closure rates [43] Possible alterations in immune cell profile [43]	Accelerated wound closure/re-epithelialization [43] Increased onset of fibroblast senescence/senescence-associated secretory phenotype (SASP) [43] Increased keratinocyte proliferation [43]
Myeloid cells	No changes in wound closure rates [44]	No changes in wound closure rates [44]

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
