# Peer review of "Regulation of Wound Healing by the NRF2 Transcription Factor—More Than Cytoprotection"

_ijms, 2019, doi:10.3390/ijms20163856_

Round 1

Reviewer 1 Report

Critical review by Dr. Werner and group elaborating NRF2's role on wound healing process which is beyond it's definitive role in cytoprotection. Few points should be discussed more broadly before it is ready for acceptance. They are as follows:

While mentioning the dark side of NRF2 in cancer, authors should shed some light on the role of AMPK and mTOR signaling in NRF2 pathways. From doi: 10.1074/jbc.M116.760249 we know that NRF2 can regulate mTOR whereas in doi: 10.1074/jbc.M114.622571 / PMCID: PMC4358122, it's been shown the role of AMPK in mTOR signaling pathways. By referring these two works authors should speculate their point of view on NRF2's role in AMPK and mTOR signaling cascade in cancer.

Continuing the above point it's been shown antidiabetic drug DPP-4i activates NRF2 pathway DOI: 10.1126/scitranslmed.aad6095 where as it has been shown in doi: 10.1080/15384101.2015.1087623/ PMCID: PMC4825547 that antidiabetic drug AICAR can be used as combination therapy with rapamycin in cancer patients. So it will be interesting to see author's perspective in this context. By citing the above mentioned works they should opine their point of view on this therapeutic angel of using antidiabetic drug in cancer patients who has high level of NRF2. 

By adding these angles I think the novelty and significance of this work/ review will be more prominent and helpful for readers to think on this perspective.

Author Response

We thank the reviewer for the valuable time invested into the review of this article and for his/her important comments and suggestions. 

The review focuses on the role of Nrf2 in wound healing, whereas the role of Nrf2 in cancer has been covered in various other excellent reviews. Therefore, we had decided not to further discuss the mechanisms involved in the pro-tumorigenic role of Nrf2. However, the suggestions of the reviewer are very important and we included some of these articles plus additional articles on mTor in wound healing into the revised version of our review (in the context of wound healing, see page 6 and page 8).

Reviewer 2 Report

This is a very well written summary of the role NRF2 plays in wound healing, with reasoned arguments and appropriate caveats. I believe it will be a useful reference text for researchers working in this area to promote wound healing via targeting of NRF2.

Author Response

We are very pleased about the positive comments of the reviewer and thank him/her for the valuable time invested into this review.